# Recombinant SARS-CoV-2 Spike Protein Stimulates Secretion of Chymase, Tryptase, and IL-1β from Human Mast Cells, Augmented by IL-33

**DOI:** 10.3390/ijms24119487

**Published:** 2023-05-30

**Authors:** Irene Tsilioni, Theoharis C. Theoharides

**Affiliations:** 1Department of Immunology, Tufts University School of Medicine, Boston, MA 02111, USA; eirini.tsilioni@tufts.edu; 2Institute of Neuro-Immune Medicine, Nova Southeastern University, Clearwater, FL 33759, USA

**Keywords:** ACE2, coronavirus, cytokines, inflammation, mast cells, spike protein, toll-like receptors

## Abstract

SARS-CoV-2 infects cells via its spike (S) protein binding to its surface receptor angiotensin-converting enzyme 2 (ACE2) and results in the production of multiple proinflammatory cytokines, especially in the lungs, leading to what is known as COVID-19. However, the cell source and the mechanism of secretion of such cytokines have not been adequately characterized. In this study, we used human cultured mast cells that are plentiful in the lungs and showed that recombinant SARS-CoV-2 full-length S protein (1–10 ng/mL), but not its receptor-binding domain (RBD), stimulates the secretion of the proinflammatory cytokine interleukin-1β (IL-1β) as well as the proteolytic enzymes chymase and tryptase. The secretion of IL-1β, chymase, and tryptase is augmented by the co-administration of interleukin-33 (IL-33) (30 ng/mL). This effect is mediated via toll-like receptor 4 (TLR4) for IL-1β and via ACE2 for chymase and tryptase. These results provide evidence that the SARS-CoV-2 S protein contributes to inflammation by stimulating mast cells through different receptors and could lead to new targeted treatment approaches.

## 1. Introduction

Severe acute respiratory syndrome coronavirus 2 (SARS-CoV-2) infection of the cells is caused by the interaction of the viral spike (S) protein receptor-binding domain (RBD) with the surface receptor, known as angiotensin-converting enzyme 2 (ACE2) [1,2]. SARS-CoV-2 infection leads to the secretion of a “storm” [3,4] of proinflammatory cytokines [3,4,5,6,7,8,9,10,11,12,13,14,15,16,17], which is closely associated with COVID-19 development [3,18]. However, the cell source and the mechanism of secretion of such cytokines have not been adequately characterized.

Mast cells are ubiquitous tissue immune cells that are involved in health and disease [19,20,21,22,23]. Mast cells are plentiful in the lungs and can secrete many proinflammatory cytokines such as interleukin-1β (IL-1β) and interleukin-6 (IL-6), as well as chemokines such as CXCL8 [24], but also the proteolytic enzymes chymase and tryptase [25,26]. Mast cells respond to viruses [27,28] and have also been implicated in COVID-19 as targets of SARS-CoV-2 [29,30,31,32,33,34,35]. Recent studies investigating autopsy material reported an increased number of degranulated mast cells in the lungs of patients with COVID-19 [36,37,38] associated with perivascular inflammation [37,39]. Of note, another paper showed significantly increased expression of mast cell-related genes, as well as an increased number of chymase and tryptase-positive mast cells in the lungs of patients with COVID-19 pneumonia as compared to other patients with influenza pneumonia [40]. However, the exact mechanism of mast cell activation and the mediators secreted are still understudied.

Here we show that SARS-CoV-2 spike protein has a synergistic action with interleukin-33 (IL-33), stimulating human mast cells to secrete the proteases chymase and tryptase, as well as the proinflammatory cytokine IL-1β, thus contributing to inflammation.

## 2. Results

### 2.1. SARS-CoV-2 Spike Protein Stimulates Secretion of Chymase, Tryptase, and IL-1β from Human Mast Cells in a Dose-Dependent Manner

Initially, we identified the optimal concentration of full-length S protein (1–10 ng/mL) that maximally stimulates the secretion of preformed mediators, such as chymase and tryptase, and proinflammatory cytokines, including IL-1β, from human cultured mast cells. Stimulation of 10 ng/mL recombinant full-length SARS-CoV-2 S protein for 1 h and 24 h, respectively, significantly increased the secretion of chymase, tryptase, and IL-1β levels in the cell culture supernatants, compared with those in controls in a dose-dependent manner (Figure 1A–C). Equivalent data were also obtained using FL-Spike protein from GeneTex (GTX136780-pro, Irvine, CA, USA), (data not shown). 

### 2.2. SARS-CoV-2 Spike Protein but Not RBD Stimulate Secretion of IL-1β from Human Mast Cells

The effects of recombinant SARS-CoV-2, full-length S, and RBD on the secretion of preformed (chymase and tryptase) and proinflammatory mediators (IL-1β) from human mast cells were further examined. Stimulation of LADR cells with 10 ng/mL of full-length S, but not RBD for 24 h, resulted in a significant release of IL-1β compared to controls (Figure 2C). Notably, both 10 ng/mL of full-length S and RBD were able to induce the release of chymase and tryptase levels (Figure 2A,B). Nonetheless, stimulation with full-length S (10 ng/mL) was more potent compared to stimulation with RBD (10 ng/mL) for 24 h. Equivalent results were also obtained using RBD protein from GeneTex (GTX136716-pro, Irvine, CA, USA), (data not shown). 

### 2.3. IL-33 Significantly Augments the Ability of SARS-CoV-2 Spike Protein to Stimulate Secretion of Chymase, Tryptase, and IL-1β from Human Mast Cells

Administration of recombinant full-length S (10 ng/mL) and IL-33 (30 ng/mL) together for 1 h or 24 h stimulates increased secretion of chymase and tryptase (Figure 3A,B) as well as IL-1β (Figure 3C) from LADR cells, respectively, compared with cells treated by recombinant full-length S alone. 

### 2.4. SARS-CoV-2 Spike Protein Stimulates Chymase and Tryptase Secretion from Human Mast Cells via ACE2

Next, we investigated if SARS-CoV-2 S-induced proinflammatory responses in human mast cells are mediated by TLR- or ACE2-dependent mechanism. We preincubated human mast cells with the following: (1) anti-TLR2 Ab, (2) anti-TLR4 Ab, and (3) anti-ACE2 Ab. LADR mast cells stimulation by full-length S (10 ng/mL) or RBD (10 ng/mL) for 1 h resulted in a significant release of chymase and tryptase, which was inhibited by pretreatment with 2 μg/mL of anti-ACE2 Ab (Figure 4A,B). Nevertheless, preincubation with anti-TLR2 or anti-TLR4 Ab did not decrease preformed mediator release (Figure 4A,B). 

### 2.5. SARS-CoV-2 Spike Protein Stimulates IL-1β Secretion from Human Mast Cells via TLR4

We further examined whether secretion of IL-1β from LADR cells after stimulation with the recombinant SARS-CoV-2 full-length S is mediated by TLR- or ACE2-signaling. LADR cells were preincubated with anti-TLR2, anti-TLR4, and anti-ACE2 antibodies and stimulated with full-length S for 24 h. Stimulation of human mast cells for 24 h by full-length S significantly increased the secretion of IL-1β, while pretreatment with anti-TLR4 Ab suppressed it (Figure 4C). However, pretreatment of LADR cells with anti-TLR2 or anti-ACE2 Ab did not have any effect on the proinflammatory mediator release (Figure 4C). 

## 3. Materials and Methods

*Human Mast Cell Culture:* Human LADR mast cells [41] (kindly supplied by Dr. A Kirshenbaum, NIH, Bethesda, MD, USA) were cultured in StemPro-34 medium (Invitrogen, Waltham, MA, USA) supplemented with 2 mM L-glutamine, 1% penicillin/streptomycin and 100 ng/mL rhSCF (Orphan Biovitrum AB, Stockholm, Sweden). The proliferation of the cells was tested by measuring total cell numbers weekly. A Trypan blue (0.4%) exclusion assay was used to determine cell viability.

*LADR Mast Cell Treatments*: LADR mast cells were stimulated with recombinant full-length SARS-CoV-2 S (1–10 ng/mL; ab281471, Abcam, Cambridge, UK) or RBD (1–10 ng/mL; ab273065, Abcam, Cambridge, UK), and/or preincubated with the following: (1) anti-TLR2 antibody (2 μg/mL, 1 h; maba2-htlr2, InvivoGen, San Diego, CA, USA), (2) anti-TLR4 antibody (2 μg/mL, 1 h; mabg-htlr4, InvivoGen, San Diego, CA, USA) and (3) anti-ACE2 antibody (2 μg/mL, 1 h; ab108209, InvivoGen, San Diego, CA, USA). The anti-TLR antibodies were used at concentrations and incubation times, as reported previously [42]. Substance P (SP) (2 μM, S6883, Sigma-Aldrich, St. Louis, MO, USA), IL-33 (30 ng/mL, 3625-IL, R&D Systems, Minneapolis, MN, USA) and their combination were used as positive controls. IL-33 was added at the same time as the full-length SARS-CoV-2 S and the cells were treated for 24 h. Moreover, cells were treated for 24 h with SP, IL-33 alone, and the combination of SP and IL-33. 

*Cell Viability Assay*: Cells were also tested at different concentrations (1, 5, and 10 ng/mL) of SARS-CoV-2 S and RBD. Viable cell numbers were counted by trypan blue (0.4%) exclusion using a hemocytometer, and viability was better than 95%.

*Proinflammatory Mediator Secretion:* LADR mast cells were seeded at 2.5 × 10^5^ cells/0.5 mL in 48-well culture plates (Becton Dickinson, Franklin Lakes, NJ, USA) prior to preincubation with anti-TLRs or anti-ACE2 ab (for 1 h). Mast cells, upon activation by various triggers, can secrete prestored mediators such as tryptase and chymase through rapid (5 min to 1 h) degranulation, while the secretion of newly synthesized cytokines, including IL-1β, requires 24 h. Therefore, for IL-1β secretion, mast cells were stimulated with recombinant full-length SARS-CoV-2 S or RBD for 24 h, while for chymase and tryptase secretion, mast cells were stimulated for 1 h. Mast cell supernatant fluids were collected and stored at −80 °C until further analysis. Concentrations of IL-1β, chymase, and tryptase were measured using commercially available manual ELISA kits from R&D Systems (Minneapolis, MN, USA) and BosterBio (Pleasanton, CA, USA). The minimum detectable level for the IL-1β was 3.91–250 pg/mL. For chymase, it was 93.8–6000 pg/mL, and for tryptase, it was 156–10,000 pg/mL.

*Statistical Analysis:* All experiments were performed in triplicate wells for each condition and were repeated at least three times. Results from cultured cells are presented as mean ± SEM. Comparisons were made between control and stimulated cells using the unpaired two-tailed t-test. In addition, multiple comparisons were made between (1) stimulated cells without inhibitors and those with inhibitors using one-way ANOVA (analysis of variance), followed by post hoc analysis by the Dunnett multiple-comparisons test; and (2) all the conditions with inhibitors among themselves using one-way ANOVA, followed by post hoc analysis by the Tukey multiple-comparisons test. All statistical analyses were performed using GraphPad Prism 9 software.

## 4. Discussion

Infection by SARS-CoV-2 involves attachment, fusion, and cellular entry facilitated by the S protein via binding with high affinity to the host cell surface receptor ACE2 [1]. In this study, we used human cultured mast cells that are plentiful in the lungs and showed that recombinant full-length SARS-CoV-2 S protein, but not the RBD, stimulates the secretion of the proinflammatory cytokine IL-1β. Interestingly, both full-length S and RBD stimulate the secretion of the proteolytic enzymes chymase and tryptase from human mast cells. A recent paper also reported that recombinant SARS-CoV-2 protein and RBD can stimulate cultured human mast cells to secrete both chymase and tryptase [43]. However, the authors of this paper used microgram amounts of S and RBD proteins, while we used nanogram amounts (1000 less).

Two papers reported elevated blood levels of chymase more than those of tryptase in patients with COVID-19 [44,45]. In addition, a recent study showed that mast cells promote viral entry via the formation of chymase–spike complexes [46], while another paper revealed that histamine potentiates SARS-CoV-2 entry into endothelial cells [47]. Both chymase and tryptase are supposedly stored in the same secretory granules of connective tissue mast cells, while only chymase is predominantly found in mucosal mast cells [48]. The *apparent* greater serum levels of chymase compared to tryptase may imply that mostly mucosal mast cells are activated in the lungs of patients with COVID-19. It is of interest that chymase is known to convert angiotensin I to angiotensin II [49] and may act in an autocrine fashion to increase the expression of ACE2, which then facilitates viral entry. Furthermore, tryptase could promote inflammation via the activation of protease-activated receptors [50], especially since it has been implicated in LPS-induced lung inflammation in mice [51] and enhances the migration of human lung fibroblasts [52].

We also show for the first time that secretion of IL-1β, chymase, and tryptase is significantly augmented by the alarmin IL-33. This result is of clinical relevance as blood IL-1β levels are increased in patients with COVID-19 [53,54], and administration of the IL-1 receptor antagonist Anakinra improved outcomes of severe COVID-19 patients [54]. IL-33 has been implicated in severe COVID-19 [55,56,57,58,59,60,61,62]. Moreover, IL-33 was reported to increase the expression of ACE2 [63]. We had reported increased IL-33 soluble receptor (ST2) in the serum of severe COVID-19 patients that we interpreted as a futile effort to neutralize increased circulating IL-33 [64]. We had also previously shown that IL-33 has a synergistic effect with the peptide SP in stimulating the secretion of impressive amounts of IL-1β from cultured human mast cells [65] without degranulation [66]. Differential secretion could be regulated by distinct regulatory mechanisms [67,68,69,70].

Secretion of IL-1β does not appear to be mediated via ACE2 since the RBD had no effect, and secretion was inhibited by pretreatment with an antibody blocking TLR-4, but not ACE2. Instead, the secretion of chymase and tryptase appeared to be mediated via ACE2 since the RBD stimulated such secretion, and it was inhibited by the ACE2 blockade. However, the mechanism via which activation of ACE2 leads to degranulation and secretion of chymase and tryptase is not presently known. SARS-CoV-2 has been reported to activate TLRs [42,71,72], leading to the secretion of immune molecules that could contribute to COVID-19 symptoms [42,73]. Moreover, activation of TLR4 increased the expression of ACE2 [74], further enhancing viral infectivity in an autocrine loop. 

Mast cells can, therefore, contribute to COVID-19 [32] and possibly also to long-COVID [39,75]. Preventing or minimizing the detrimental effects of the spike protein could lead to novel targeted treatment approaches [76,77,78]. In particular, secretion of IL-1β from mast cells could be inhibited by the natural flavonoids luteolin and tetrametho-xyluteolin [65], while the actions of IL-1β could be blocked by the biologic Anakinra [55]. However, our study has some limitations. The use of primary human mast cells should also be considered. Moreover, in all the experimental settings, a cytofluorimetric analysis should be performed to monitor the expression of the ACE2 receptor, TLR, and IL-33 receptor before and after the different stimuli, as well as after the addition of neutralizing antibodies.

## Figures and Tables

**Figure 1 ijms-24-09487-f001:**
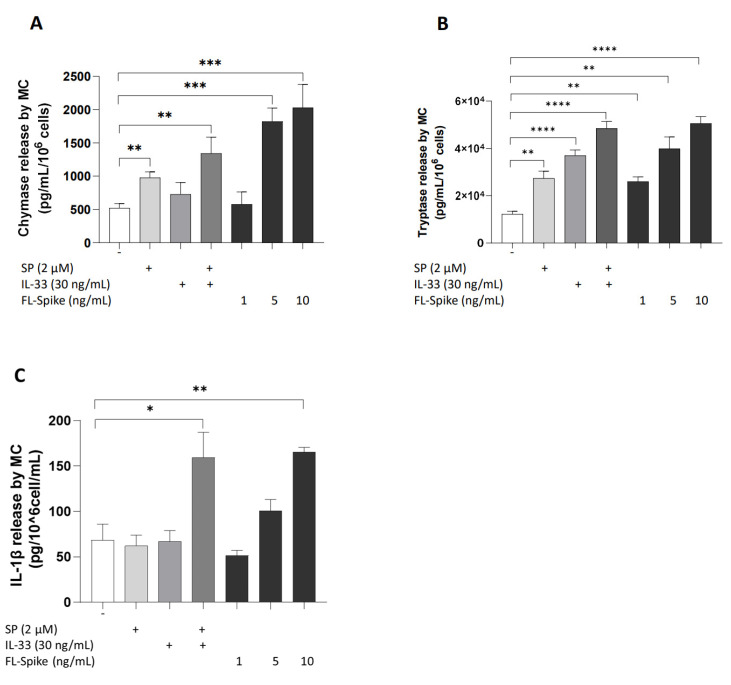
**SARS-CoV-2 spike protein stimulates the secretion of proteases and IL-1β from human mast cells in a dose-dependent manner**. LADR mast cells (2.5 × 10^5^ cells) were stimulated with recombinant full-length SARS-CoV-2 S protein (1–10 ng/mL) for 1 h (**A**,**B**) and 24 h (**C**). Secretion of chymase (**A**), tryptase (**B**), and IL-1β (**C**) was determined by specific ELISAs. SP (2 μM), IL-33 (30 ng/mL), and their combination were used as “positive” controls. All conditions were performed in triplicate for each dataset and repeated 3 times (*n* = 3). The horizontal brackets indicate the corresponding levels of significance when present (*p* < 0.05 (*), *p* < 0.01 (**), *p* < 0.001 (***), and *p* < 0.0001 (****)).

**Figure 2 ijms-24-09487-f002:**
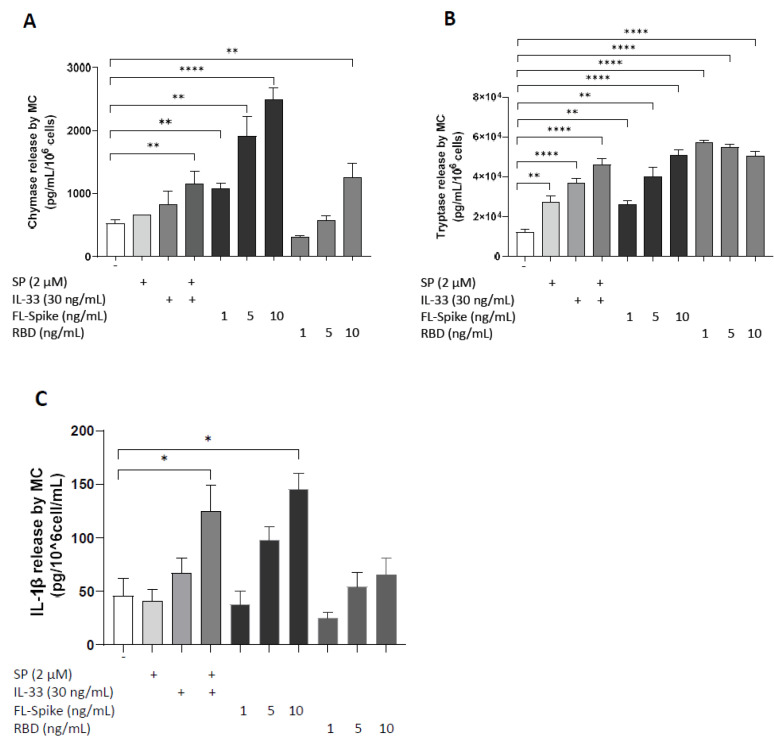
**SARS-CoV-2 spike but not RBD stimulates secretion of IL-1β from human mast cells.** LADR mast cells (2.5 × 10^5^ cells) were stimulated with recombinant full-length SARS-CoV-2 S protein (1–10 ng/mL) and RBD (1–10 ng/mL) for 1 h (**A**,**B**) and 24 h (**C**). Secretion of chymase (**A**), tryptase (**B**), and IL-1β (**C**) was determined by specific ELISAs. SP (2 μM), IL-33 (30 ng/mL), and their combination were used as “positive” controls. All conditions were performed in triplicate for each dataset and repeated 3 times (*n* = 3). The horizontal brackets indicate the corresponding levels of significance when present (*p* < 0.05 (*), *p* < 0.01 (**), and *p* < 0.0001 (****)).

**Figure 3 ijms-24-09487-f003:**
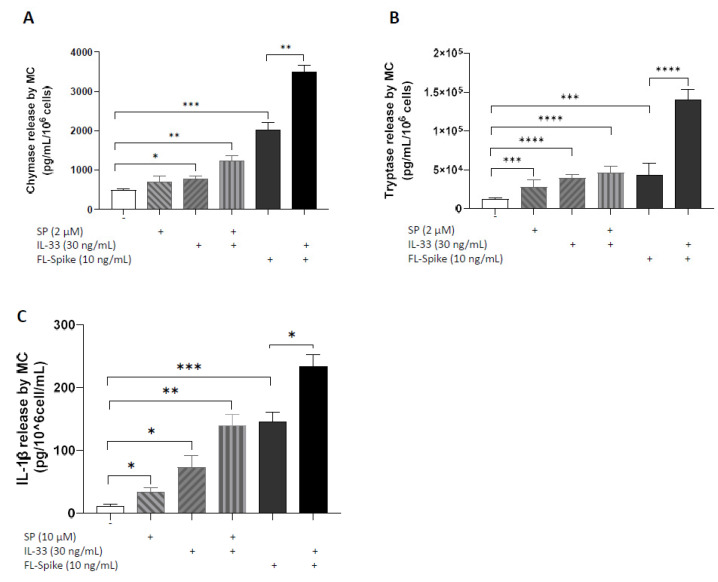
**IL-33 significantly augments the ability of SARS-CoV-2 spike protein to stimulate the secretion of chymase, tryptase, and IL-1β from human mast cells**. LADR mast cells (2.5 × 10^5^ cells) were stimulated with recombinant full-length SARS-CoV-2 S protein (10 ng/mL) and IL-33 (30 ng/mL) for 1 h (**A**,**B**) and 24 h (**C**). Secretion of chymase (**A**), tryptase (**B**), and IL-1β (**C**) was determined by specific ELISAs. SP (2 μM), IL-33 (30 ng/mL), and their combination were used as “positive” controls. All conditions were performed in triplicate for each dataset and repeated 3 times (*n* = 3). The horizontal brackets indicate the corresponding levels of significance when present (*p* < 0.05 (*), *p* < 0.01 (**), *p* < 0.001 (***), and *p* < 0.0001 (****)).

**Figure 4 ijms-24-09487-f004:**
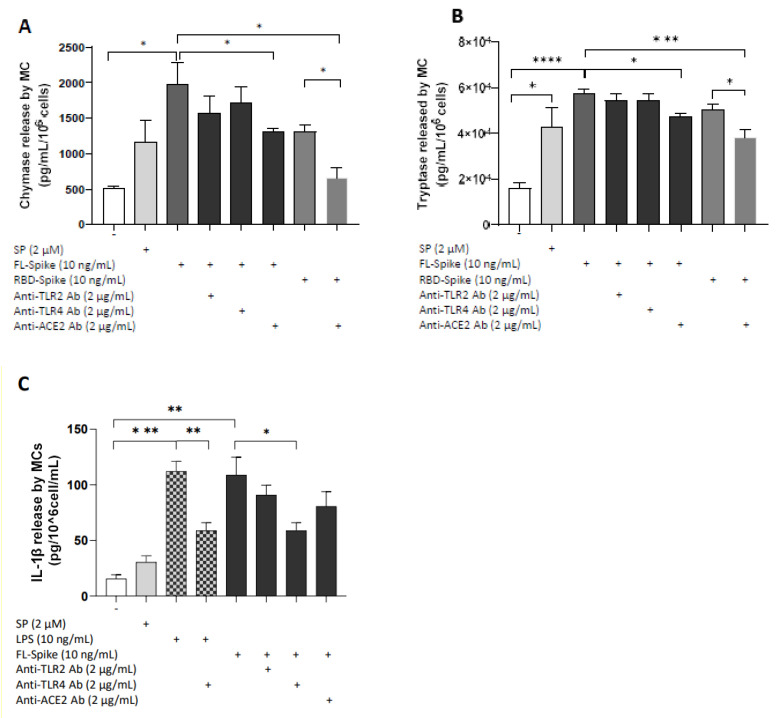
**Full-length SARS-CoV-2 spike stimulates IL-1β secretion from human mast cells via TLR4 signaling.** LADR mast cells (2.5 × 10^5^ cells) were pretreated with anti-TLR2 Ab (2 μg/mL), anti-TLR4 Ab (2 μg/mL), or anti-ACE2 Ab (2 μg/mL) for 1 h and then stimulated with recombinant full-length SARS-CoV-2 S (10 ng/mL) for 1 h and 24 h, respectively. Secretion of chymase (**A**), tryptase (**B**)**,** and IL-1β (**C**) was determined by specific ELISAs. All conditions were performed in triplicate for each dataset and repeated 3 times (*n* = 3). The horizontal brackets indicate the corresponding levels of significance when present (*p* < 0.05 (*), *p* < 0.01 (**), *p* < 0.001 (***), and *p* < 0.0001 (****)).

## Data Availability

Data available upon request.

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
