# Peer review of "Recombinant SARS-CoV-2 Spike Protein Stimulates Secretion of Chymase, Tryptase, and IL-1β from Human Mast Cells, Augmented by IL-33"

_ijms, 2023, doi:10.3390/ijms24119487_

Round 1

Reviewer 1 Report

It is important to discover the role of pro-inflammatory, inflammatory and receptor proteins in the COVID-19 development. These experiments are very interesting. Major knowledge in this issue can contribute to design and test novel pharmacological molecules to treat the severe respiratory disease by SARS-CoV-2 infection.

Some minor revisions are required:

Line 61: It must be extended the abbreviation IL only one for all mentions and SP (peptide substance P).

-Line 66-67 : Why were there choosen different times, 24h vs 1h, for the stimulation of IL-1β or chymase/tryptase secretion? Please you, to justify this choice.

-Lines 69: It is important to refer additional data about ELISA testing? Manual or automathic? Specimens were processed immediately  after the sampling or following an established freezing storage.

-Line 72: The conditions performed in triplicate could be extended briefly.

-Line 76: One-way analysis of variance can be abbreviated.

Lines 84-87: Can you add some aspects about the identification of optimal S-protein concentration? I.e. experiments, references. I have noted the different concentrations in the figure 1. It must be added these data in the main body.

Line 95-96: Please you to revise the correct time of stimulation. At line 96, the term "respectively" is refer to? This concept is hazy.

Figure 1-4: Please you to justify in the figure legend the significance of the different number of asterisks.

-For the Discussion section, it could be evaluated with an overview on pubblished references the control of IL-1β secretion in the pharmacological treatment.

Author Response

Manuscript ijms-2289581

Response to Reviewers

Dear Editor,

Thank you for giving us the opportunity to submit a revised draft of the manuscript “Recombinant SARS-CoV-2 Spike Protein stimulates secretion of Chymase, Tryptase and IL-1b from human mast cells, augmented by IL-33” for publication in the International Journal of Molecular Sciences. We appreciate the time and effort that you and the reviewers dedicated to providing feedback on our manuscript and are grateful for the insightful comments on and valuable improvements to our paper. We have incorporated most of the suggestions made by the reviewers. Those changes are highlighted within the manuscript. We want to note that this is a “Short Communication” paper. Therefore, we could not possibly perform all the additional important experiments suggested for lack of sufficient funding presently. However, these additional experiments have been added to the conclusion as work that should be done in the future. Please see below, in blue, for a point-by-point response to the reviewers’ comments and concerns.

______________________________________________________________________________

Reviewer 1

Comment #1: Line 61: It must be extended the abbreviation IL only one for all mentions and SP (peptide substance P).

Author Response: Thank you for pointing this out. This correction has been made accordingly.

Comment #2: Line 66-67 : Why were there choosen different times, 24h vs 1h, for the stimulation of IL-1β or chymase/tryptase secretion? Please you, justify this choice.

Author Response: Thank you for your comment. Mast cells upon activation by various triggers can secrete prestored mediators such as tryptase and chymase through rapid (5min to 1hr) degranulation, while the secretion of newly synthesized cytokines, including interleukin-1β (IL-1β) requires 24 hr. This explanation has now been added to the Materials and Methods section.

Comment #3: Lines 69: It is important to refer additional data about ELISA testing? Manual or automatic? Specimens were processed immediately  after the sampling or following an established freezing storage.

Author Response: This is an excellent suggestion. As recommended by the reviewer, we have now included information regarding the ELISA testing, as well as the specimens processing.

Comment #4: Line 72: The conditions performed in triplicate could be extended briefly.

Author Response: Thank you for your comment. The conditions have now been expanded as necessary.

Comment #5: Line 76: One-way analysis of variance can be abbreviated.

Author Response: Thank you for your comment. This correction has been made accordingly.

Comment #6: Lines 84-87: Can you add some aspects about the identification of optimal S-protein concentration? I.e., experiments, references. I have noted the different concentrations in the figure 1. It must be added these data in the main body.

Author Response: Thank you. We have added the data in the Results section.

Comment #7: Line 95-96: Please you to revise the correct time of stimulation. At line 96, the term "respectively" is refer to? This concept is hazy.

Author Response: Thank you for this comment. As suggested by the reviewer, we have now revised the correct time of stimulation and made the appropriate corrections.

Comment #8: Figure 1-4: Please you to justify in the figure legend the significance of the different number of asterisks.

Author Response: Thank you for this comment. This correction has been made accordingly.

Comment #9: For the Discussion section, it could be evaluated with an overview on published references the control of IL-1β secretion in the pharmacological treatment.

Author Response: We had avoided discussing possible treatments because this is a “Brief Communication”. However, we have now added a paragraph mentioning some natural and medicinal approaches to regulating IL-1β secretion and action.

­­­­­­­­­­­­­­­­­­­­­­­_____________________________________________________________________

Manuscript ijms-2289581

Response to Reviewers

Dear Editor,

Thank you for giving us the opportunity to submit a revised draft of the manuscript “Recombinant SARS-CoV-2 Spike Protein stimulates secretion of Chymase, Tryptase and IL-1b from human mast cells, augmented by IL-33” for publication in the International Journal of Molecular Sciences. We appreciate the time and effort that you and the reviewers dedicated to providing feedback on our manuscript and are grateful for the insightful comments on and valuable improvements to our paper. We have incorporated most of the suggestions made by the reviewers. Those changes are highlighted within the manuscript. We want to note that this is a “Short Communication” paper. Therefore, we could not possibly perform all the additional important experiments suggested for lack of sufficient funding presently. However, these additional experiments have been added to the conclusion as work that should be done in the future. Please see below, in blue, for a point-by-point response to the reviewers’ comments and concerns.

______________________________________________________________________________

Reviewer 1

Comment #1: Line 61: It must be extended the abbreviation IL only one for all mentions and SP (peptide substance P).

Author Response: Thank you for pointing this out. This correction has been made accordingly.

Comment #2: Line 66-67 : Why were there choosen different times, 24h vs 1h, for the stimulation of IL-1β or chymase/tryptase secretion? Please you, justify this choice.

Author Response: Thank you for your comment. Mast cells upon activation by various triggers can secrete prestored mediators such as tryptase and chymase through rapid (5min to 1hr) degranulation, while the secretion of newly synthesized cytokines, including interleukin-1β (IL-1β) requires 24 hr. This explanation has now been added to the Materials and Methods section.

Comment #3: Lines 69: It is important to refer additional data about ELISA testing? Manual or automatic? Specimens were processed immediately  after the sampling or following an established freezing storage.

Author Response: This is an excellent suggestion. As recommended by the reviewer, we have now included information regarding the ELISA testing, as well as the specimens processing.

Comment #4: Line 72: The conditions performed in triplicate could be extended briefly.

Author Response: Thank you for your comment. The conditions have now been expanded as necessary.

Comment #5: Line 76: One-way analysis of variance can be abbreviated.

Author Response: Thank you for your comment. This correction has been made accordingly.

Comment #6: Lines 84-87: Can you add some aspects about the identification of optimal S-protein concentration? I.e., experiments, references. I have noted the different concentrations in the figure 1. It must be added these data in the main body.

Author Response: Thank you. We have added the data in the Results section.

Comment #7: Line 95-96: Please you to revise the correct time of stimulation. At line 96, the term "respectively" is refer to? This concept is hazy.

Author Response: Thank you for this comment. As suggested by the reviewer, we have now revised the correct time of stimulation and made the appropriate corrections.

Comment #8: Figure 1-4: Please you to justify in the figure legend the significance of the different number of asterisks.

Author Response: Thank you for this comment. This correction has been made accordingly.

Comment #9: For the Discussion section, it could be evaluated with an overview on published references the control of IL-1β secretion in the pharmacological treatment.

Author Response: We had avoided discussing possible treatments because this is a “Brief Communication”. However, we have now added a paragraph mentioning some natural and medicinal approaches to regulating IL-1β secretion and action.

Reviewer 2 Report

The paper by Tsilioni & Theoharides first investigates the ability of several ng/mL concentrations of full length and RBD of the SARS-CoV-2 spike protein to elicit release of chymase, tryptase and IL-1B from the mast cell line LADR. The authors then identify that the alarmin IL-33 increases the release of these proteins when mast cells are exposed to full length spike protein. Finally, the authors test the ability of pre-incubation with several antibodies to inhibit these effects to elucidate which extracellular receptors are involved. Overall, the manuscript builds on an observation covered only briefly in previous research, that SARS-CoV-2 spike protein can elicit the release of proteases and signaling molecules. Given the potential importance of mast cells in acute and long CoVID-19, this research has important implications for understanding and treating disease caused by SARS-CoV-2.

Major Comments

1) The authors seem a bit vague about the mechanism by which tryptase and chymase are released. Given the short time frame (~1hr), is degranulation not the most likely mechanism? The authors should comment on whether they think degranulation is involved in the discussion section. Have the authors performed B-Hexosaminidase assays with cells exposed to spike protein? This would be a significantly cheaper and quicker way at looking at degranulation than ELISA if degranulation is indeed elicited by spike protein.

2) Have the authors tested the effect of SARS-CoV-2 spike and RBD on LADR viability? If not, the authors should perform an assay where LADR cells are exposed to the tested concentrations of SARS-CoV-2 spike (1, 5 and 10ng/mL) for 1 hour and membrane integrity is measured by trypan blue assay or viability dye. Additionally, controls exposed to only an equal amount of whatever buffer the spike/RBD protein is stored in should be performed alongside these assays. This will ensure that any effects are not due to the buffer in which the spike protein is stored. These experiments will demonstrate that the spike protein is not simply causing lysis of the cells, releasing chymase and tryptase.

3) For figure 4, positive controls should be offered for inhibitors to show they are working and what extent of inhibition can be expected (eg. pretreat cells with anti-TLR2 antibody then treat with TLR2 agonist for 1hr and compare release vs cells exposed to no antibody).

4) The authors should provide justifications and citations (unless stated by the manufacturer) for the antibodies they chose to inhibit TLR2, TLR4 and ACE2. Are these specific antibodies known to inhibit these receptors or target them for internalization/degradation?

5) More experiments will be needed to reach the conclusion that "SARS-CoV-2 Spike protein stimulates chymase and tryptase secretion from human mast cells via ACE2" and "SARS-CoV-2 Spike protein stimulates IL-1β secretion from human mast cells via TLR4". For TLR4, the most straightforward way would be to look at phosphorylation of downstream signaling molecules by western blot. For ACE2, the authors should provide a proposed mechanism/signaling pathway for how binding of spike protein to ACE2 induces release of chymase and tryptase and provide at least one additional experiment to test this hypothesis. If a mechanism cannot be found, the authors may try alternative means to confirm ACE2 is involved. If cells can be transfected, siRNAs may be used to downregulate ACE2. Alternatively, if there is a spike protein mutant that can be obtained that doesn't bind ACE2 this could also be used.

Minor Comments

6) In figure 1 & 2, the authors write "for 1 h and 24 h, respectively" (line 96 and 113) but it is unclear what respectively refers to. Perhaps this could be rewritten to make clear which subfigures they are referring to, eg. " for 1 h (A & B) or 24 h (C)".

7) The authors should provide the product number for the SARS-CoV-2 S and RBD from Abcam. It seems there are many different products offered so this will be necessary to narrow down which specific spike/RBD is being used.

8) It is unclear in the materials and methods how Figure 3 was conducted. The authors should add somewhere in lines 63-67 when IL-33 was added when combined with spike protein. Specifically, was it added at the same time as the spike protein, or were cells pre-incubated with IL-33. The authors should also indicate, in samples treated with SP, IL-33 alone and SP + IL-33 how long they treated cells with each of these for.

9) Figure 4B seems to be missing a + for the last column (presumably to show that the anti-ACE2 ab is used).

Author Response

Manuscript ijms-2289581

Response to Reviewers

Dear Editor,

Thank you for giving us the opportunity to submit a revised draft of the manuscript “Recombinant SARS-CoV-2 Spike Protein stimulates secretion of Chymase, Tryptase and IL-1b from human mast cells, augmented by IL-33” for publication in the International Journal of Molecular Sciences. We appreciate the time and effort that you and the reviewers dedicated to providing feedback on our manuscript and are grateful for the insightful comments on and valuable improvements to our paper. We have incorporated most of the suggestions made by the reviewers. Those changes are highlighted within the manuscript. We want to note that this is a “Short Communication” paper. Therefore, we could not possibly perform all the additional important experiments suggested for lack of sufficient funding presently. However, these additional experiments have been added to the conclusion as work that should be done in the future. Please see below, in blue, for a point-by-point response to the reviewers’ comments and concerns.

Reviewer 2

Comment #1: The authors seem a bit vague about the mechanism by which tryptase and chymase are released. Given the short time frame (~1hr), is degranulation not the most likely mechanism? The authors should comment on whether they think degranulation is involved in the discussion section. Have the authors performed B-Hexosaminidase assays with cells exposed to spike protein? This would be a significantly cheaper and quicker way at looking at degranulation than ELISA if degranulation is indeed elicited by spike protein.

Author Response: Thank you for the comment. Although beta-hexosaminidase would be cheaper, we preferred to measure tryptase and chymase because they are stored together with beta-hexosaminidase in the same secretory granules and have clinical relevance.

Comment#2: Have the authors tested the effect of SARS-CoV-2 spike and RBD on LADR viability? If not, the authors should perform an assay where LADR cells are exposed to the tested concentrations of SARS-CoV-2 spike (1, 5 and 10ng/mL) for 1 hour and membrane integrity is measured by trypan blue assay or viability dye. Additionally, controls exposed to only an equal amount of whatever buffer the spike/RBD protein is stored in should be performed alongside these assays. This will ensure that any effects are not due to the buffer in which the spike protein is stored. These experiments will demonstrate that the spike protein is not simply causing lysis of the cells, releasing chymase and tryptase.

Author Response: Thank you for this comment. We have tested the different concentrations (1, 5 and 10 ng/mL) of SARS-CoV-2 spike and RBD on cell viability by trypan blue assay, and viability was better than 95%. Description of these experiments is in the Material and Methods section.

Comment #3: For figure 4, positive controls should be offered for inhibitors to show they are working and what extent of inhibition can be expected (e.g., pretreat cells with anti-TLR2 antibody then treat with TLR2 agonist for 1hr and compare release vs cells exposed to no antibody).

Author Response: Thank you for this comment. As a positive control for TLR4 involvement, we used LPS, which is known to activate TLR4  on cell lines and mouse models. 

Comment #4: The authors should provide justifications and citations (unless stated by the manufacturer) for the antibodies they chose to inhibit TLR2, TLR4 and ACE2. Are these specific antibodies known to inhibit these receptors or target them for internalization/degradation?

Author Response: Thank you for this comment. Previous studies showed that SARS-CoV-2 can activate monocytes and macrophages through interaction with TLR2, TLR4 and ACE2 on cell lines and models. Please see the publications below:

  • Dhriti Kaushika , Ranjana Bhandari and Anurag Kuhad. TLR4 as a therapeutic target for respiratory and neurological complications of SARS-CoV-2. Expert Opin Ther Targets 2021;25(6):491-508. PMID:33857397
  • Kyung Mok Sohn , Sung-Gwon Lee , Hyeon Ji Kim et al. COVID-19 Patients Upregulate Toll-like Receptor 4-mediated Inflammatory Signaling That Mimics Bacterial Sepsis. J Korean Med Sci. 2020;35(38):e343. PMID:32989935

Comment #5: More experiments will be needed to reach the conclusion that "SARS-CoV-2 Spike protein stimulates chymase and tryptase secretion from human mast cells via ACE2" and "SARS-CoV-2 Spike protein stimulates IL-1β secretion from human mast cells via TLR4". For TLR4, the most straightforward way would be to look at phosphorylation of downstream signaling molecules by western blot. For ACE2, the authors should provide a proposed mechanism/signaling pathway for how binding of spike protein to ACE2 induces release of chymase and tryptase and provide at least one additional experiment to test this hypothesis. If a mechanism cannot be found, the authors may try alternative means to confirm ACE2 is involved. If cells can be transfected, siRNAs may be used to downregulate ACE2. Alternatively, if there is a spike protein mutant that can be obtained that doesn't bind ACE2 this could also be used.

Author Response: Thank you for this comment. We have now included LPS as a positive control for TLR4 stimulation. Since this is a short communication, we have not investigated signal transduction pathways. Moreover, we would like to remind the reviewer that additional information that ACE2 does not work is provided by the lack of any effect of RBD.

Comment #6: In figure 1 & 2, the authors write "for 1 h and 24 h, respectively" (line 96 and 113) but it is unclear what respectively refers to. Perhaps this could be rewritten to make clear which subfigures they are referring to, eg. " for 1 h (A & B) or 24 h (C)".

Author Response: Thank you! This correction has been made accordingly.

Comment #7: The authors should provide the product number for the SARS-CoV-2 S and RBD from Abcam. It seems there are many different products offered so this will be necessary to narrow down which specific spike/RBD is being used.

Author Response: Thank you! This information has been added to the Materials and Methods section.

Comment #8: It is unclear in the materials and methods how Figure 3 was conducted. The authors should add somewhere in lines 63-67 when IL-33 was added when combined with spike protein. Specifically, was it added at the same time as the spike protein, or were cells pre-incubated with IL-33. The authors should also indicate, in samples treated with SP, IL-33 alone and SP + IL-33 how long they treated cells with each of these for.

Author Response: Thank you for your comment. This information has been added to the Materials and Methods section.

Comment #9: Figure 4B seems to be missing a + for the last column (presumably to show that the anti-ACE2 ab is used).

Author Response: Thank you for bringing this to our attention. This information has been added to Figure 4B.

_____________________________________________________________________

Reviewer 3 Report

In their manuscript, the authors present interesting results on the capacity of SARS-CoV-2 spike protein and RBD on the activation of mast cells and the subsequent release of chymase, tryptase and IL-1b, and show additionally augmentation by the alarmin IL-33. Although the manuscript reads sound, a major critisism has to be raised that should be addressed by the authors.

Major comment:

Figure 4 and corresponding result section: The heading and content of this section do not fit with the actual results. The authors descripe stimulation chymase and tryptase secretion by SARS-CoV-2 spike protein via ACE2, however, this mechanism seems only to be true for tryptase but not chymase, since figure 4A doesn't show inhibition by anti-ACE2 for chymase. In fact, there is inhibition of anti-ACE2 on the release of chymase and tryptase stimulated by RBD, but these results are not described. Furhter, statstical significance for RBD and inhibition by anti-ACE2 in Fig. 4A is missing. Also, an asterix indicating anti-ACE incubation on RBD in Fig. 4B is missing. Since the inhibition of anti-ACE2, as compared to negaitve control, is not complete, i.e. potentially other mechanisms involved, the specificity of antibody inhibition could be questionable. Therefore, isotype controls for anti-ACE2 Ab would be recommended.

Minor comments:

Authors should indicate the use of abbreviations for SP and FL-Spike either in the corresponding figure legends or in general.

Authors should indicate the origine of SP and IL-33 in materials and methods.

Author Response

Manuscript ijms-2289581

Response to Reviewers

Dear Editor,

Thank you for giving us the opportunity to submit a revised draft of the manuscript “Recombinant SARS-CoV-2 Spike Protein stimulates secretion of Chymase, Tryptase and IL-1b from human mast cells, augmented by IL-33” for publication in the International Journal of Molecular Sciences. We appreciate the time and effort that you and the reviewers dedicated to providing feedback on our manuscript and are grateful for the insightful comments on and valuable improvements to our paper. We have incorporated most of the suggestions made by the reviewers. Those changes are highlighted within the manuscript. We want to note that this is a “Short Communication” paper. Therefore, we could not possibly perform all the additional important experiments suggested for lack of sufficient funding presently. However, these additional experiments have been added to the conclusion as work that should be done in the future. Please see below, in blue, for a point-by-point response to the reviewers’ comments and concerns.

Reviewer 3

Comment #1: Figure 4 and corresponding result section: The heading and content of this section do not fit with the actual results. The authors describe stimulation chymase and tryptase secretion by SARS-CoV-2 spike protein via ACE2, however, this mechanism seems only to be true for tryptase but not chymase, since figure 4A doesn't show inhibition by anti-ACE2 for chymase. In fact, there is inhibition of anti-ACE2 on the release of chymase and tryptase stimulated by RBD, but these results are not described. Further, statistical significance for RBD and inhibition by anti-ACE2 in Fig. 4A is missing. Also, an asterix indicating anti-ACE incubation on RBD in Fig. 4B is missing. Since the inhibition of anti-ACE2, as compared to negative control, is not complete, i.e. potentially other mechanisms involved, the specificity of antibody inhibition could be questionable. Therefore, isotype controls for anti-ACE2 Ab would be recommended.

Author Response: Thank you for these comments. These corrections have been made accordingly for Figure 4A and Figure 4B. 

Comment #2: Authors should indicate the use of abbreviations for SP and FL-Spike either in the corresponding figure legends or in general.

Author Response: Thank you for this comment. We have now used abbreviations for SP and Full-length Spike protein.

Comment #3: Authors should indicate the origin of SP and IL-33 in materials and methods.

Author Response: Thank you for your comment. This information has been added to the Materials and Methods section.

Reviewer 4 Report

The manuscript by Tsilioni I et al. provides evidence that SARS-CoV-2 Spike stimulates human mast cell secretion of proteases and IL-1 beta which increase upon co-administration of IL-33. The work complements older results by another group previously showing that SARS-CoV-2 triggers mast cell release of chymase and tryptase and also of pro-inflammatory cytokines including IL-1 beta, inducing lung injury (ref 42 of this paper, doi: 10.1038/s41392-021-00849-0). The original data are related to the additive effect of IL-33, but lack some controls and need further investigation.

Major points:

My primary concern is that in all the experimental setting the effect of recombinant SARS-Cov-2 Spike protein and IL-33 has been investigated using a mast cell line. Regardless of the source, the use of primary human mast cells should be also considered.

Moreover, in all the experimental setting cytofluorimetric analysis must be performed to monitor the expression of ACE 2 receptor, TLR and IL-33 receptor before and after the different stimuli as well as after the addition of neutralizing antibodies. Of note, it has been previously reported that IL-33 stimulation increases ACE2 receptor expression on keratinocytes, as also commented by the authors (ref. 62, lines 188 and 189), and this possible interesting crosstalk should be further investigated also on mast cells.

Finally, in order to establish a possible contribution of ACE2 receptor in protease and IL-1beta production additional important controls are missing (see specific comments below).

Specific comments:

Figure 2A and B: RBD stimulates the same production of tryptase than FL-spike (panel B) while RBD is unable to stimulate chymase production. The authors should explain and discuss this result that does not fit with previous finding (ref 42).

Figure 3: IL-33 alone stimulates the production of chymase and protease but not of IL-1 beta (in panel C the slight increase appears to be not significant). However, the co-stimulation with IL-33 and FL-spike (24 hours) shows an additive effect on IL-1 beta production respect to the response in the presence of both stimuli used alone. Since IL-33 could increase ACE2 receptor expression, FACS is required to evaluate and compare the surface expression of this receptor under the different stimuli.

The authors should also make a specific comment on the synergistic (?) effect on tryptase production observed upon stimulation with IL-33 and FL-Spike (panel B).

Figure 4 and B: In panel A the inhibition of chymase production by FL-spike appears not to be significant upon the addition of anti-ACE2 Ab, however in panel B the inhibition of tryptase release is significant. LADR cells have log-fold higher granular expression of tryptase than chimase, but are the different proteases stored in different granules?

The authors should clarify this point and correct the result section, accordingly (see lines 133 and 137-139).

Figure 4C: In order to conclude that “pretreatment of anti-ACE 2 Ab did not have any effect on IL1 release (lines 158 and 159)” additional controls are required. In particular, IL1 production should be also assessed after pretreatment with anti-ACE2 Ab upon RBD (at different concentrations) and IL-33 stimulation. FACS analysis is required for this set of experiment and must be included.

The manuscript needs some editing for clarity:

The full name of the acronym SP should be indicated in the material and methods section and in figure legends.

Please verify and correct the title of figure 4 (human microglia?).

In regard to the statistical analysis, multiple comparisons were made in all the figures and the test used should be clearly indicated in each figure legend.

Author Response

Manuscript ijms-2289581

Response to Reviewers

Dear Editor,

Thank you for giving us the opportunity to submit a revised draft of the manuscript “Recombinant SARS-CoV-2 Spike Protein stimulates secretion of Chymase, Tryptase and IL-1b from human mast cells, augmented by IL-33” for publication in the International Journal of Molecular Sciences. We appreciate the time and effort that you and the reviewers dedicated to providing feedback on our manuscript and are grateful for the insightful comments on and valuable improvements to our paper. We have incorporated most of the suggestions made by the reviewers. Those changes are highlighted within the manuscript. We want to note that this is a “Short Communication” paper. Therefore, we could not possibly perform all the additional important experiments suggested for lack of sufficient funding presently. However, these additional experiments have been added to the conclusion as work that should be done in the future. Please see below, in blue, for a point-by-point response to the reviewers’ comments and concerns.

Reviewer 4

Comment #1: My primary concern is that in all the experimental settings the effect of recombinant SARSCov-2 Spike protein and IL-33 has been investigated using a mast cell line. Regardless of the source, the use of primary human mast cells should also be considered. Moreover, in all the experimental setting cytofluorimetric analysis must be performed to monitor the expression of ACE 2 receptor, TLR and IL-33 receptor before and after the different stimuli as well as after the addition of neutralizing antibodies. Of note, it has been previously reported that IL-33 stimulation increases ACE2 receptor expression on keratinocytes, as also commented by the authors (ref. 62, lines 188 and 189), and this possible interesting crosstalk should be further investigated also on mast cells. Finally, in order to establish a possible contribution of ACE2 receptor in protease and IL1beta production additional important controls are missing (see specific comments below).

Author Response: Thank you. There are important experiments that are outside the scope of a “Brief Communication” and there is no more available funding. However, they have been added in the Conclusion section.

Comment #2: Figure 2A and B: RBD stimulates the same production of tryptase than FL-spike (panel B) while RBD is unable to stimulate chymase production. The authors should explain and discuss this result that does not fit with previous finding (ref 42).

Author Response: Thank you for this comment. As indicated in the Discussion, the authors of reference #42 used 1,000 more (μΜ) Spike protein than what we used (nM), hence stimulation in #42 was pharmacological and unlikely to be physiological.

Comment #3: Figure 3: IL-33 alone stimulates the production of chymase and protease but not of IL-1 beta (in panel C the slight increase appears to be not significant). However, the co-stimulation with IL-33 and FL-spike (24 hours) shows an additive effect on IL-1 beta production respect to the response in the presence of both stimuli used alone. Since IL-33 could increase ACE2 receptor expression, FACS is required to evaluate and compare the surface expression of this receptor under the different stimuli. The authors should also make a specific comment on the synergistic (?) effect on tryptase production observed upon stimulation with IL-33 and FL-Spike (panel B).

Author Response: We thank the reviewer for this comment. We have now incorporated all the corrections for Figure A and B into the results section of our paper accordingly. 

Comment #4: Figure 4 and B: In panel A the inhibition of chymase production by FL-spike appears not to be significant upon the addition of anti-ACE2 Ab, however in panel B the inhibition of tryptase release is significant. LADR cells have log-fold higher granular expression of tryptase than chymase, but are the different proteases stored in different granules? The authors should clarify this point and correct the result section, accordingly (see lines 133 and 137-139).

Author Response: Thank you for this comment. We have now incorporated all the corrections for Figure A and B into the results section of our paper accordingly.  

Comment #5: Figure 4C: In order to conclude that “pretreatment of anti-ACE 2 Ab did not have any effect on IL1 release (lines 158 and 159)” additional controls are required. In particular, IL1 production should be also assessed after pretreatment with anti-ACE2 Ab upon RBD (at different concentrations) and IL-33 stimulation. FACS analysis is required for this set of experiments and must be included.

Author Response: Thank you for this suggestion. However, we have not done the experiment and we don’t have more funding to conduct it.

Comment #6: The manuscript needs some editing for clarity: The full name of the acronym SP should be indicated in the material and methods section and in figure legends. Please verify and correct the title of figure 4 (human microglia?). In regard to the statistical analysis, multiple comparisons were made in all the figures and the test used should be clearly indicated in each figure legend.

Author Response: Thank you for this comment. We have now incorporated all the suggested corrections on our paper accordingly.

Round 2

Reviewer 2 Report

I believe the authors have sufficiently answered the comments I provided. 

Reviewer 4 Report

The methods are now adequately detailed and some results better presented.